# TMD Design by an Entropy Index for Seismic Control of Tall Shear-Bending Buildings

**DOI:** 10.3390/e25081110

**Published:** 2023-07-25

**Authors:** Yumei Wang, Zhe Qu

**Affiliations:** 1Earthquake Engineering Research & Test Center, Guangzhou University, Guangzhou 510006, China; 2Institute of Engineering Mechanics, China Earthquake Administration, Harbin 150080, China; quz@iem.ac.cn

**Keywords:** shear-bending building, Grammians, Hankel singular values (HSVs), information entropy, optimal TMD

## Abstract

This study proposes a new arrangement-tuning method to maximize the potential of tuned mass dampers (TMDs) in decreasing the seismic responses of tall buildings. The method relies on a Grammian-based entropy index with the physical meaning of covariance responses to white noise without the involvement of external inputs. A twelve-story RC frame-shear wall building was used as an example to illustrate the method. Indices were computed for the building with TMDs placed on different stories and tuning to different modes and were compared with responses to white noise (colored) time histories. Results showed that greater index reduction cases agree well with greater story-drift reductions cases, despite the differences in the time step of the white noises and structural model types (pure shear vs. shear-bending), and the optimal TMD is not necessarily the traditional “roof—1st mode tuning” case. Comparisons were also made for the shear-bending building under seven earthquake excitations. It is found that, though TMDs are not full-band effective controllers, the index-selected TMDs still perform the best in three out of seven earthquakes. So, the proposed internal-property-based entropy index provides a good controller design for large-scale structures under unpredictable none-stationary excitations.

## 1. Introduction

Tuned mass dampers (TMDs) are widely used in the vibration control of buildings. Their efficiency is influenced by many factors, such as the TMD arrangement, TMD parameters, excitation characteristics, etc. Traditionally, the roof and the 1st mode is the standard TMD configuration for the seismic control of buildings. E.g., Elias S. and Matsagar V.’s study showed the effectiveness of this TMD scheme on a benchmark building [1]. Wu [2] developed a placement procedure and conducted numerical and experimental studies. Daniel and Lavan [3] presented an optimal TMD allocation design methodology for an irregular building. Rahmani et al. [4] also studied TMD placement using a genetic algorithm. These studies showed that the optimal locations of TMD correspond to the nodes of lower modes and earthquake-sensitive modes. These results agree with common sense; however, energy exchange does not always occur at tuning frequencies. Extensive investigations have indicated that TMDs are less effective in seismic control than wind control of high-rise buildings. One of the reasons is that this roof/1st mode scheme is more controllable to the neighboring stories for the load distribution as of earthquakes [2]. So, more attention was paid to multiple-TMD schemes for seismic control [5,6]. Another important reason is that mode-related indices, such as the modal participation coefficient, have apparent limitations because they emphasise some response contributions while ignoring others, which can be viewed as error controls on specified structural degrees of freedom (DOFs) [7].

On the other hand, lateral loads will cause both shear and bending in buildings. However, seismic design and analysis are usually based on shear models because the error from neglecting bending is insignificant for most low buildings. But some buildings, for example, frame-shear wall buildings, have synergistic shear and bending deformation. In these cases, ignoring bending deformation would lead to an incorrect evaluation of dynamic behaviors and, thus, may invalidate the seismic control measures. But so far, only a few studies can be found specifically on bending influences on seismic performance, i.e.,: Kaneko [8] employed a lumped mass stick model with shear springs and rotational springs to take into account the overall bending deformation of a structure, and then deduced an optimal damping and equivalent stiffness of the TMD. Ping et al. [9] integrated an optimum TMD into the shear-bending building and performed different control methods to examine the control efficiency.

To design an “optimal” controller (TMD) for earthquake resistance of a shear-bending building, the first important thing is to find an appropriate model. This study would adopt the lumped-mass shear-bending model constructed by Wada et al. [10], in which a stiffness matrix is modified to account for the bending effects for shear-bending buildings. Pei et al. [11] used Wada’s model to study the bending effects of tall shear-bending structures. Another important thing is to find a criterion or a standard that could measure the controller’s contribution to the structural system; then, a model that could account for the bending effects is required, or else it would result in wrong simulations, invalidate the criterion, and adversely amplify the responses. Physical quantities to characterize the contribution of structural properties should have characteristics of [7]: (1) dimensionless; (2) invariant with coordinate transformation and have a unique definite value; (3) could be normalized so that the sum be 1. As stated above, some modal quantities agree with the requirements, but their limitations and deficiencies are also prominent.

Due to the importance of locations, control contribution is also examined by controller placement procedures. Most of these methods involve optimization techniques, genetic algorithms, modal objective functions, etc. However, studies also showed that some norm-based properties, like controllability and observability Grammians, were also suitable for placement problems. Grammians contain second-order system information and reflect state correlations and energy in structural feedback control. So, it can also be shaped for the evaluation of structural performance. Kang and Shin [12] determined the accelerometer locations for bridges using the frequency-domain Hankel matrix, which is related to the eigenvalues of the cross-Grammians. Bigoni et al. [13] proposed to use the variational approximation of sparse Gaussian processes to place a fixed number of sensors over a structure of interest.

This study aims to refine the TMD design with different placement and tuning combinations for tall buildings with bending effects. The design schemes and control effects will be evaluated by the Grammian-based index. This norm-based index had also been constructed into some “information entropy” to be used as a performance measure for model reduction (Fu et al. [14,15]). This study would interpret the connotation of the index given entropy and disclose how it reflects the structural internal properties, and thus, could measure the contribution of TMDs through value change with location and tuning and help to determine the optimal TMD design.

## 2. Methods

### 2.1. Shear-Bending Lumped Mass Model

The deformation of a shear-bending building consists of two parts: shear deformation and bending deformation. In the lumped mass model, corresponding story stiffnesses are represented by shear and bending springs, respectively. The schematic of the lumped mass model and spring-deformation relationships are shown in Figure 1 [10].

In Figure 1, $Q_{i}$ is story shear force, $\delta_{i}$ is the shear displacement, $R_{i}$ is bending moment, $\varphi_{i}$ is bending rotation angle, and $u_{i}$ is the lateral displacement. The stiffness of the story shear spring $k_{0i}$ is the shear force $Q_{i}$ when the story shear offset $\delta_{i}=1$, and the story stiffness of the bending spring $k_{b}$ is the bending moment $R_{i}$ when the story bending rotation angle $\varphi_{i}=1$. For general structural components (columns or walls), $k_{0}$ and $k_{b}$ can be obtained by Equation (1):(1)$$k_{0}=\frac{12EI}{h^{3}}\;(\mathrm{fixed}\;\mathrm{ends},\;\mathrm{middle}\;\mathrm{stories}),\;k_{0}=\frac{3EI}{h^{3}}\;(\mathrm{one}\;\mathrm{end}\;\mathrm{free},\;\mathrm{roof}),\;k_{b}=\frac{R}{\varphi }=\frac{EI}{h}$$
where E is the modulus of elasticity, I is the moment of inertia and h is the story height. The overall shear-bending stiffness is the resultant of the shear stiffness and bending stiffness in parallel, as in Equation (2):(2)$$k_{s}=\frac{1}{h/GA+1/k_{0}}=\frac{k_{0}}{1+\phi }$$
where G is the shear modulus, A is the area of the cross-section, ϕ is the ratio of section bending stiffness and shear stiffness, and
(3)$$\phi =\frac{3EI/h^{3}}{GA/H}=\frac{3EI}{GAh^{2}}\;(\mathrm{roof}),\;\phi =\frac{12EI/h^{3}}{GA/H}=\frac{12EI}{GAh^{2}}(\mathrm{middle}\;\mathrm{stories})$$

For the ith story, the force-displacement relationship is ${\left\{\begin{array}{cc} \begin{array}{cc} X_{i} & M_{i} \end{array} & \begin{array}{cc} X_{i+1} & M_{i+1} \end{array} \end{array}\right\}}^{T}={K_{i}\left\{\begin{array}{cc} \begin{array}{cc} u_{i} & \theta_{i} \end{array} & \begin{array}{cc} u_{i+1} & \theta_{i+1} \end{array} \end{array}\right\}}^{T}$, in which the stiffness matrix $K_{i}$ (middle stories) is shown in Equation (4):(4)$$K_{i}=\left[\begin{array}{cc} \begin{array}{cc} k_{si} & \frac{{k_{si}h}_{i}}{2}\\ \frac{{k_{si}h}_{i}}{2} & \frac{{k_{si}h}_{i}^{2}}{4}+k_{b} \end{array} & \begin{array}{cc} -k_{si} & \frac{{k_{si}h}_{i}}{2}\\ -\frac{{k_{si}h}_{i}}{2} & \frac{{k_{si}h}_{i}^{2}}{4}-k_{b} \end{array}\\ \begin{array}{cc} -k_{si} & -\frac{{k_{si}h}_{i}}{2}\\ \frac{{k_{si}h}_{i}}{2} & \frac{k_{si}h_{i}^{2}}{4}-k_{b} \end{array} & \begin{array}{cc} k_{si} & -\frac{{k_{si}h}_{i}}{2}\\ -\frac{k_{si}h_{i}}{2} & \frac{{k_{si}h}_{i}^{2}}{4}+k_{b} \end{array} \end{array}\right]\;=\frac{EI}{1+\phi_{i}}\left[\begin{array}{cc} \begin{array}{cc} \frac{12}{h_{i}^{3}} & \frac{6}{h_{i}^{2}}\\ \frac{6}{h_{i}^{2}} & \frac{4+\phi_{i}}{h_{i}} \end{array} & \begin{array}{cc} -\frac{12}{h_{i}^{3}} & \frac{6}{h_{i}^{2}}\\ -\frac{6}{h_{i}^{2}} & \frac{2-\phi_{i}}{h_{i}} \end{array}\\ \begin{array}{cc} -\frac{12}{h_{i}^{3}} & -\frac{6}{h_{i}^{2}}\\ \frac{6}{h_{i}^{2}} & \frac{2-\phi_{i}}{h_{i}} \end{array} & \begin{array}{cc} \frac{12}{h_{i}^{3}} & -\frac{6}{h_{i}^{2}}\\ -\frac{6}{h_{i}^{2}} & \frac{4+\phi_{i}}{h_{i}} \end{array} \end{array}\right]$$

In shear structures, $k_{b}≌0$ and ϕ≌0. So with respect to shear structures, the overall story stiffness of shear-bending structures decreases by $\frac{1}{1+\phi }$ times, and the rotation components increase by $\frac{\phi_{i}}{h_{i}}$ in the near end and decrease by $\frac{\phi_{i}}{h_{i}}$ in the far end.

Then, the shear-bending lumped mass model of the entire building is:(5)$$\left[\begin{array}{cc} M & 0\\ 0 & J \end{array}\right]\left\{\begin{array}{c} \ddot{y}\\ \ddot{\theta } \end{array}\right\}+C\left\{\begin{array}{c} \dot{y}\\ \dot{\theta } \end{array}\right\}+\left[\begin{array}{cc} K^{yy} & K^{y\theta }\\ K^{\theta y} & K^{\theta \theta } \end{array}\right]\left\{\begin{array}{c} y\\ \theta \end{array}\right\}=-\left[\begin{array}{cc} M & 0\\ 0 & J \end{array}\right]\left\{\begin{array}{c} 1\\ 0 \end{array}\right\}{\ddot{a}}_{g}$$
where y and θ are displacements and rotation angle concerning the ground, ${\ddot{x}}_{g}$ is ground acceleration. J is the rotation moment of inertia. The stiffness matrices of $K^{yy}$, $K^{\theta \theta }$, $K^{y\theta }=K^{\theta y}$ are synthesized from the story stiffness matrix $K_{i}$, and are rearranged by the u_i_ and θ_i_. C is the damping matrix.

In this study, the damping matrix is assumed to be second-order Rayleigh damping, i.e.,
(6)$$C=\frac{2\zeta }{{(\omega }_{1}+\omega_{2})(\omega_{1}\omega_{2}M+K)}$$
where $\omega_{1}$ and $\omega_{2}$ are the first two frequencies of the structure, ζ is the damping ratio. Here ζ= 5% for all DOFs, which is common for reinforced concrete structures. The 5% damping could be neglected in the condensation process of Equations (7)–(9).

J and M are usually not in the same order. To make computations converge faster, J is divided by $H^{2}$ (H is the sum of story heights, i.e., $H=\sum_{0}^{n}h_{i}$) and rewrite Equation (4) into Equation (7):(7)$$\left[\begin{array}{cc} M & 0\\ 0 & \frac{J}{H^{2}} \end{array}\right]\left\{\begin{array}{c} \ddot{y}\\ H\ddot{\theta } \end{array}\right\}+\left[\begin{array}{cc} K^{yy} & \frac{K^{y\theta }}{H}\\ \frac{K^{\theta y}}{H} & \frac{K^{\theta \theta }}{H^{2}} \end{array}\right]\left\{\begin{array}{c} y\\ H\theta \end{array}\right\}=-\left[\begin{array}{cc} M & 0\\ 0 & \frac{J}{H^{2}} \end{array}\right]\left\{\begin{array}{c} 1\\ 0 \end{array}\right\}{\ddot{a}}_{g}$$

If the rotational components of earthquakes are not considered, $J/H^{2}=0$, Equation (7) then could be condensed. Substitute $J/H^{2}=0$ into the 2nd equation of Equation (7) yields:(8)$$\left\{H\theta \right\}=-{\left[\frac{K^{\theta \theta }}{H^{2}}\right]}^{-1}\left[\frac{K^{\theta y}}{H}\right]\left\{y\right\}$$

Then, substituting Equation (8) into the 1st equation of Equation (7) yields Equation (9):(9)$$\left[M\right]\ddot{\left\{y\right\}}+{\left[K^{yy}\right]}^{*}\left\{y\right\}=-\left[M\right]\left\{1\right\}{\ddot{a}}_{g},\;\mathrm{where}\;{\left[K^{yy}\right]}^{*}=\left[K^{yy}\right]-\left[\frac{K^{y\theta }}{H}\right]{\left[\frac{K^{\theta \theta }}{H^{2}}\right]}^{-1}\left[\frac{K^{\theta y}}{H}\right]$$

Equations (8) and (9) both contain n (the story numbers) equations, and $\left\{H\theta \right\}$ can be deduced from $\left\{y\right\}$. The statically condensed equation of the shear-bending model thus has the same number of DOFs as the shear model.

When TMDs add to the system, just expand the $\left[M\right]$ and ${\left[K^{yy}\right]}^{*}$ by adding the DOF of the TMDs and modify the terms in ${\left[K^{yy}\right]}^{*}$ that couple with the TMD. i.e., just add $\left[\begin{array}{ccc} \begin{array}{cc} 0 & \ldots \end{array} & \begin{array}{cc} {-\omega }^{2}m_{T} & \begin{array}{cc} \ldots & 0 \end{array} \end{array} & \omega^{2}m_{T} \end{array}\right]$ to the last line of ${\left[K^{yy}\right]}^{*}$, where the ith term, ${-\omega }^{2}m_{T}$ and the (n+1)^th^ term, $\omega^{2}m_{T}$ (ω is the tuning frequency), construct the recovery force from the relative motion of the TMD to its installed story. At the same time, the ith line of ${\left[K^{yy}\right]}^{*}$ should also consider the retroaction of the TMD.

### 2.2. HSV-Based Entropy Index

#### 2.2.1. The Entropy Based on Hankel Singular Values (HSVs)

Let $x={\left[\begin{array}{cc} y & \dot{y} \end{array}\right]}^{T}$, u be the external input (for earthquakes, **u** =$\;M\left\{1\right\}{\ddot{a}}_{g}$ as in Equation (9), the equations of motion and responses of a system can be written in the state-space form of:(10)$$\left\{\begin{array}{c} \dot{x}=Ax+Bu\\ z=C_{out}x+D_{out}u \end{array}\right.,\;\mathrm{where}\;A=\left[\begin{array}{cc} 0 & I\\ {-M}^{-1}K & {-M}^{-1}C \end{array}\right]$$
where **z** is the output, A is the state matrix, C is the damping matrix, and $B,C_{out},D_{out}$ are input coefficient, output gain, and output coefficient matrices, respectively.

The influence of the location of “external” forces, such as earthquakes and active control forces, is included in the input gain, i.e., the **B** matrix; while TMD force is essentially “internal”, because it is the direct function of structural responses, and thus, its location impact is on the mass and/or the stiffness matrix, in all, the **A** matrix.

The controllability Grammian $W_{c}$ and observability Grammian $W_{o}$ are defined as
(11)$$\int_{0}^{\infty }e^{-At}BB^{T}e^{-A^{T}t}dt=\underset{t\to \infty }{\lim}E\left\{x(t)x^{T}(t)\right\}.\;W_{o}=\int_{0}^{\infty }e^{-A^{T}t}C_{out}^{T}C_{out}e^{-At}dt=\underset{t\to \infty }{\lim}E\left\{\bar{x}(t){\bar{x}}^{T}(t)\right\}.$$
where e is the base of the natural logarithm, $E\left\{.\right\}$ represents mathematical expectation. For continuous stable systems, $W_{c}$ is the steady-state covariance of the states of the system $\left(A,\;B\right)$, and $W_{o}$ is the steady-state covariance of the states of the dual system $\left(A^{T},C_{out}^{T}\right)$, when u is a zero-mean Gaussian white noise [16].

Fernado et al. [17] defined the cross-Grammian matrix for an asymptotically stable and continuous system:(12)$$G_{cross}=\sqrt{W_{c}W_{o}}=\underset{t\to \infty }{\lim}\sqrt{E\left\{x(t)x^{T}(t)\right\}E\left\{\bar{x}(t){\bar{x}}^{T}(t)\right\}}$$

For a stable controllable and observable system, the cross-Grammian $G_{cross}$ and the Hankel singular value (HSV), are internally related, as Equation (13) shows:(13)$$\gamma =\sqrt{{\lambda (W}_{c}W_{o})}$$
where λ(.) denotes the eigenvalue.

Not that, for a system, $W_{c}$ is constant, while $W_{o}$ varies with output gains: displacement, velocity, acceleration, recovery forces, etc. have different $C_{out}$. Also, $W_{c}$ and $W_{o}$ vary with coordinate systems, but γ are invariant with coordinate transformation and are independent of input. So γ is an index of the internal property of a system.

Fu et al. [14] defined an information entropy by cross-Grammian, which is:(14)$$I\left(G_{cross}\right)=\frac{n}{2}\log\left(2\pi e\right)+\frac{1}{2}\log\left(det\left(G_{cross}\right)\right)=\frac{n}{2}\log\left(2\pi e\right)+\frac{1}{2}\log\left(\prod_{1}^{n}\gamma_{i}\right)$$
where in modal coordinate, n is the modal number. Notice that
(15)$$\log\left(\prod_{1}^{n}\gamma_{i}\right)=\sum_{1}^{n}(log\gamma_{i})$$

$\prod_{1}^{n}\gamma_{i}$ is the determinant of HSVs; $\sum_{1}^{n}\left(log\gamma_{i}\right)$ is the logarithm of sum HSVs. So, the 2nd term of $I\left(G_{cross}\right)$ could reflect the eigenvalue of Grammians of the whole system and each order.

However, this entropy definition has deficiencies: (1) Adding the two terms together is not a good idea: the 1st term $\frac{n}{2}\log\left(2\pi e\right)$ increases linearly with modal order n; while in the 2nd term, the importance of modes ($\gamma_{i}$) decreases with modal order. (2) HSV values of different responses have a different order in magnitudes. For example, displacements or drifts are usually in the negative order of 10th, while accelerations are $\omega^{2}$ times the displacements. After taken the logarithm, acceleration $log\gamma_{i}$ may be able to keep positive, but displacement $log\gamma_{i}$ may be negative and keep decreasing when considering more modes. As a result, the summation of the logarithm $\sum_{1}^{n}\left(log\gamma_{i}\right)$ may increase for accelerations and decrease for displacement with order increases, which does not make sense. In all, it does not meet the requirements of an index. So, it has to be re-constructed for this problem.

To combine the second-order (energy) Grammian-based conception $\gamma_{i}$ with the physically meaningful property of modes, it is necessary to transfer Equation (10) to the modal coordinate equation, and then construct a normalized and stable index to evaluate the TMD contribution based on modal $\gamma_{i}$.

#### 2.2.2. Vibration Reduction Evaluation of TMDs

It turns out that, for each mode, the Hankel singular values (HSV) with a set of controllers or sensors is the root mean square (RMS) sum of the HSVs with every single controller or sensor from this set, i.e., $\gamma_{i}=\sqrt{\sum_{j=1}^{s}\gamma_{ij}^{2}}$. This property provides a means to normalize the indices using Hankel norms so that the indices are between 0 and 1. The normalized index $\sigma_{ij}$ that evaluates the ith device at the jth mode can be arranged into an s × n matrix σ in terms of HSVs defined as [16]:(16)$$\sigma_{ij}={\left\|G_{ij}\right\|}_{h}/{\left\|G\right\|}_{h}={{(\gamma }_{j})}_{i}/2\gamma_{max}$$

The σ could be examined either by column vector (Equation (17)) or by row vector (Equation (18)). Using a column vector, the kth entry is the importance index of the kth device to all modes, while using a row vector, the kth entry is the importance of all devices to a single mode. Column vector, $\sigma_{device}$, is what this study is interested in to characterize the entropy of the system with TMDs.
(17)$$\sigma_{device}={\left[\begin{array}{cc} \begin{array}{cc} \sigma_{1d} & \sigma_{2d} \end{array} & \begin{array}{cc} \ldots & \sigma_{nd} \end{array} \end{array}\right]}^{T},\;\mathrm{where}\;\sigma_{kd}=\sum_{i=1}^{n}\sigma_{ik}$$
(18)$$\sigma_{mode}=\left[\begin{array}{cc} \begin{array}{cc} \sigma_{m1} & \sigma_{m2} \end{array} & \begin{array}{cc} \ldots & \sigma_{ms} \end{array} \end{array}\right],\;\mathrm{where}\;\sigma_{mk}=\sqrt{\sum_{j=1}^{s}\sigma_{ij}^{2}}=\gamma_{i}/\gamma_{max}$$

Because HSV values are invariant (${\left\|G\right\|}_{h}$ is constant), and usually are larger in lower modes. So, with all the properties and advantages shown above, σ is a proper entropy index to evaluate the impact of the TMD on structures. This study would adopt row vector $\sigma_{kd}=\sum_{i=1}^{n}{{(\gamma }_{i})}_{k}$, and emphasize on first several modes of inter-story drifts and story acceleration HSVs.

## 3. Building Information

The building for the case study is a 12-story frame-shear wall building described in reference [11]. The height of the 1st story is 4.5 m, and the other story is 3.6 m. The building model is shown in Figure 2.

Story masses of the building are as follows: the 1st story: 1649.67 kN/m^2^; roof: 1299.97 kN/m^2^; other stories: 1579.73 kN/m^2^. Pei et al. [11] obtained the stiffness of the frame part and the shear-wall part, respectively, by pushover analysis in Midas, in which the story shear (V)-drift (u) relationships of the frame and the wall were obtained by weakening the walls and the frame manually in turn, and the M-θ relationships were obtained from story shear V and rotation angle θ. The stiffness of the frames and the shear walls are denoted by k_0_ and k_s,_ respectively. The values are shown in Table 1.

The first three natural periods of the shear-bending structures are 0.978 s, 0.361 s, 0.222 s. The modal nodes are at stories 12th, 6th, 9th + 4th, respectively. So, intuitively, the first trial is to place the TMDs on the modal node stories/tuning to that mode, written as: 12th/1st mode, 6th/2nd mode, 9th/3rd mode, and 4th/3rd modes. Let the TMD mass be three times the roof mass, i.e., $m_{T}=2106\times 10^{3}$ kN is about 5.95% of the whole structure + TMD, and the TMD‘s damping ratio be $\zeta_{T}=$ 15%.

Because the physical meaning of the index σ is the normalized covariance modal responses to white noise, the performance of these TMDs is first examined by subjecting the structure to Gaussian white noise ground motion, with its magnitude generated by wgn (4000,1,1,1) in Matlab (4000 points, power of 1 dB Watt across 1 ohm). The time step of which is set to be 0.01s for this case. The maximum story responses (displacements, drifts, accelerations) of the four TMD-controlled cases and the uncontrolled structures are shown in Figure 3.

Figure 3 shows that the 12th/1st mode TMD is good in controlling the overall displacements and accelerations, but for controlling higher inter-story drifts, the 6th/2nd mode TMD is better. The other two TMD-controlled cases are even worse than the uncontrolled case. Then, the question arose: are the 12th/1st mode and the 6th/2nd mode cases the best schemes? Are they equally good for both white noise and earthquake excitations? Are there other better combinations? So, next, the HSV reduction ratios (controlled/uncontrolled) are used to direct the TMD design.

## 4. HSV Indices of the Structure with and without TMD

### 4.1. Entropy Index Ratios

Here, optimal TMDs refer to those leading to greater HSV entropy reductions in lower modes. The procedures to apply the entropy index in optimal location/tuning determination are:

(1)Transfer Equation (10) into a modal form, and then calculate grammians $W_{c}$ and $W_{o}$.(2)Calculate γ by Equation (13), and then the index $\sigma_{u}$ of the uncontrolled building (without TMD) by Equation (18), including inter-story drifts $\sigma_{u}$ and acceleration $\sigma_{u}$.(3)Calculate $\sigma_{ij}$ of the controlled building with TMD tuning to the first three modes and placing on stories 4–12 in turn, to get a 3 × 12 matrix, as the flow chart is shown in Figure 4.(4)Calculate HSV reduction ratio $\sigma_{ij}./\sigma_{u}$ of each tuning case, and then plot the reduction ratios vs. modes as in Figure 5, Figure 6 and Figure 7, respectively.(5)Figure out the TMD location case with maximum index reduction for each tuning.

The index reduction shows that, the 12th/1st mode case is not the most index reduction case. For example, to reduce the drift HSVs, the best cases seem to be the 4th/1st mode, the 6th/2nd mode, or the 8th/3rd mode. Acceleration HSVs and drift HSVs are generally similar.

Remember that the physical meaning of HSVs is the covariance to zero-mean Gaussian white noise. So, the 4th/1st mode, 6th/2nd mode, and 8th/3rd mode are “optimal” in energy reduction under white noise. Also, the index could not tell what modes the inputs excite the most, i.e., the index could not consider the influence of frequency components of earthquakes.

### 4.2. Validation by Responses to White Noise Excitation

As is known to all, TMDs are frequency-sensitive passive controllers. They have a narrow effective frequency band. However, HSVs, or Grammians, are based on the system’s inherent properties, not on external input properties, while all frequencies, in and out of the band, would uniformly amplify the HSV-selected TMD impacts on structures. As a result, even with the placement design, TMD’s validity still rely on the earthquake type. So, using white noise to examine our method is a good start.

Figure 3 shows the responses of structures with TMDs tuning to the mode and placing at modal node stories under Gaussian white noise, in which the 12th/1st mode case looks the best among the four cases. This section compares the “optimal” TMD performance with the 12th/1st mode case under the same white noise. The maximum story responses are shown in Figure 8, in which the 12th/1st mode is the red star line:

Figure 8 shows that the “optimal” TMDs lead to greater response reduction than the 12th/1st mode case in this white noise, no matter in controlling the displacement, inter-story drift, or accelerations. Among these, the 4th/1st mode and the 8th/3rd mode cases better control the drifts, while the 8th/3rd mode case controls the accelerations. Again, the 12th/1st mode case is not very good at controlling higher story responses. Frequency responses confirm the conclusions, as shown in Figure 9.

Time history responses of the three TMD cases, the 12th/1st mode case, and the uncontrolled cases to the above white noise are shown in Figure 10.

The “optimal” TMDs perform better than the 12th/1st mode case. Though the reduction is insignificant at a few peaks, the overall reduction is apparent. So, the Grammian-based entropy can reflect the energy information of the structure influenced by TMDs.

### 4.3. Comparison with the Shear Structure Model

The shear structure refers to the structure in Figure 2 and Table 1 without considering bending stiffness *k_b_*. The first three natural periods of the shear structure are 0.818, 0.324, and 0.207. The modal nodes are at stories 12th, 6th, and 9th & 4th. But by the index σ, the optimal TMDs should be 4th/1st mode, 8th/2nd mode, and 8th/3rd modes, which are different from those in shear-bending structure.

Let the three “optimal” TMDs of the shear structural model be subjected to white noise. It is found that these TMDs are not working for the shear structure for the white noise with time step Δt = 0.01: they even amplify the structure responses. However, Δt = 0.005 s works. The maximum responses of the shear structure to the Δt = 0.005 white noise are shown in Figure 11. Again, the 4th/1st mode case is the best scheme.

But for the shear-bending structural model, the previous “optimal” TMD schemes also work for Δt = 0.005 s. The maximum responses to Δt = 0.005 s white noise are shown in Figure 12. The difference from the Figure 8 (Δt = 0.01 s) is that the 6th/2nd mode case performs even better.

The time-step difference indicates the frequency difference. Though neither Δt = 0.005 s or Δt = 0.01 s falls within the resonant (or tuning) frequency of the TMDs, the above white noise simulations still show that, different “optimal” TMDs correspond to ground excitations with different dominant frequencies. However, the entropy-selected TMDs lead to mitigated seismic responses than the uncontrolled case. The fact shows that designing a passive controller for non-stationary or colored excitations, using the system’s inherent property as an index, is not a bad choice.

To see how bending effects influence the TMD design and control, the response profiles of the shear-bending structure tuning to the shear-bending frequencies and shear frequencies, respectively, are compared under Δt = 0.005 s white noise ground motions. The results are shown in Figure 13.

Figure 13 shows that, whether the TMD is on the 4th story or the 12th story, TMDs considering bending effects all perform better than not considering bending; whether bending effects are considered or not, the 4th/1st mode cases all perform better than the 12th/1st mode case. So, ignoring the bending effects would decrease the TMD control efficiency, and it is not the 12th/1st mode but the 4th/1st mode TMD controls this shear-bending structure the best under white noise excitations. The TMDs selected by the index show some advantages compared to the traditional “modal node-modal tuning” TMDs in different white noise excitations.

However, this entropy reflects the location impacts on the gain matrices and is only in the covariance/RMS sense to white noise. Its applicability to earthquake excitations is required to be tested.

## 5. Earthquake Responses of TMD Controlled Shear-Bending Structure

The effectiveness of the above optimal TMDs would be examined to the following seven earthquakes: ChiChi_WNT_SN, Kobe_KJM_EW, Taft_#1095_SE, ElCentro_ImperialValley_EW, WenChuan, Sylmar_360, and Newhall. The names include the earthquake, station, and recording directions, in which Sylmar and Newhall are two different stations of the 1994 Northridge earthquake. The earthquakes contain fault normal and fault parrell components. In this simulation, their magnitudes are scaled to PGA = 0.3 g, i.e., 2.94 m/s^2^, corresponding to the Intensity 7 fortification level in Chinese code.

Response spectra are good indicators of the seismic performance of SDOF structures. The acceleration and displacement response spectra of the seven earthquake ground motions are shown in Figure 14, in which the three-dotted vertical lines are where the first three natural periods of the shear-bending structure are located.

These earthquakes have near-fault earthquake characteristics, i.e., they all contain rich frequency components and have long-period spikes.

With the previously determined “optimal” TMDs, the structure was subjected to seven earthquakes. Time history analysis summed up the maximum story displacement reduction ratios of the controlled/uncontrolled structures of the 12th/1st mode, 4th/1st mode, 6th/2nd mode, and 8th/3rd mode cases, shown in Figure 15.

Under these earthquakes, the TMDs do not perform as well as under white noise: the 12th/1st mode TMD can keep the reduction ratio less than 1, except for the Sylmar earthquake. The roof displacement reduction ratios are listed in Table 2.

Table 2 shows that the 4th/1st mode, 6th/2nd mode, and 8th/3rd mode are effective in 4, 1, and 3 earthquakes, respectively, in which three optimal controls occur in the 4th/1st mode case, and the other four occur in the 12th/1st mode case. Taking ChiChi earthquake response as an example, the roof displacement time histories of the first 30 s are shown in Figure 16.

It can be seen that, though the peak reductions in the 4th/1st mode case seems a little larger than those in the traditional 12th/1st mode case (actually, are mostly comparable), its overall performance could betterin the sense of total energy. Because HSV entropy is a second-order quantity, it is reasonable to examine the TMD’s efficiency by Root Mean Square (RMS) responses as well, which are given in Table 3: for example, in Chichi earthquake, the RMS response is 0.9542 in the 4th/1st mode case, while it is 0.9634 in the 12th/1st mode case. Actually, three “optimal” RMS controls occur in the entropy-based TMDs, while the 12th/1st mode case has four optimal RMS controls.

From the above studies, it can be seen that, putting the TMD on the roof and tuning to the lst mode is not always the best TMD design for tall shear-bending buildings. At least for white noise excitation, the entropy-based TMDs perform better. The seven earthquakes do not represent all types of earthquakes, and for frequency sensitive controller like TMDs, there is still a nearly 50% probability that optimal TMDs are not those intuitive “modal node-modal tuning” type.

## 6. Conclusions

This study proposed a Grammian-based index to direct the TMD design. It is a second-order quantity that reflects the internal correlations of a system under white noises and thus could account for the TMD impact on structures. By sweeping the stories and tuning frequencies for a 12-storey shear-bending building, the index finally found the optimal design of the three best TMD location/tuning combinations. Contrary to traditional roof/1st mode design, it is found that the optimal TMDs were the 4th/1st mode, 6th/2nd mode, and 8th/3rd mode cases. Time history analysis results validated the better control of these TMDs than the roof/1st mode case under white noise and under three out of seven earthquakes; the index also showed the inverse effects of wrong TMDs if models not considered the bending impact; in addition to peak reduction, root mean square reduction is also essential given energy and the physical meaning of the index. The results show that the default TMD setting roof/1st mode combination is not always the optimal scheme; for random earthquakes with broad-band white noise characteristics, the proposed entropy could provide more efficient TMD design options.

## Figures and Tables

**Figure 1 entropy-25-01110-f001:**
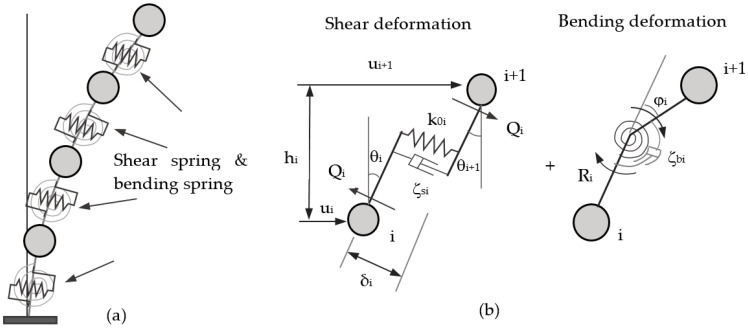
Schemes of the shear-bending lumped mass model of the multi-story building; (**a**) Deformed shape with shear & bending springs. (**b**) Separating shear and bending deformation between the ith and (i + 1)th lumped masses.

**Figure 2 entropy-25-01110-f002:**
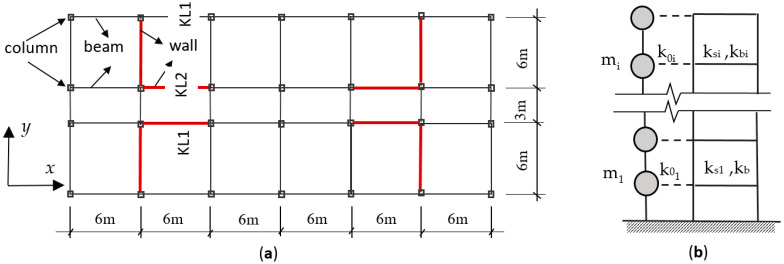
The 12-story frame-shear wall building. (**a**) Plan; (**b**) The lumped mass model: a column + shear wall (shear-bending coupled).

**Figure 3 entropy-25-01110-f003:**
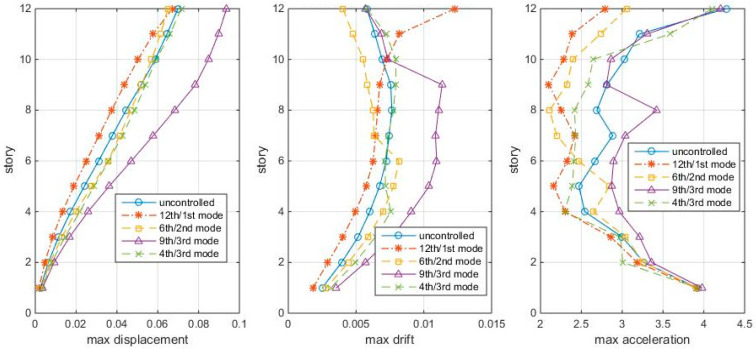
Max. story responses to white noise, with TMD at modal nodes.

**Figure 4 entropy-25-01110-f004:**
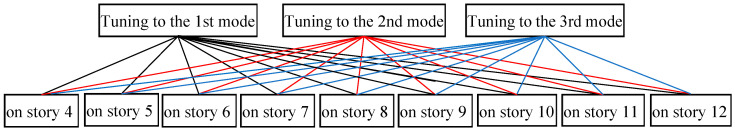
HSV-based entropy index for the building with different cases of TMDs.

**Figure 5 entropy-25-01110-f005:**
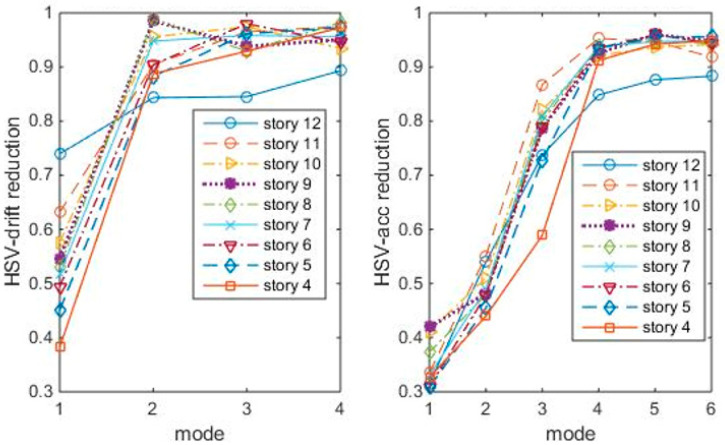
HSV reduction ratios with TMDs tuning to the 1st modes.

**Figure 6 entropy-25-01110-f006:**
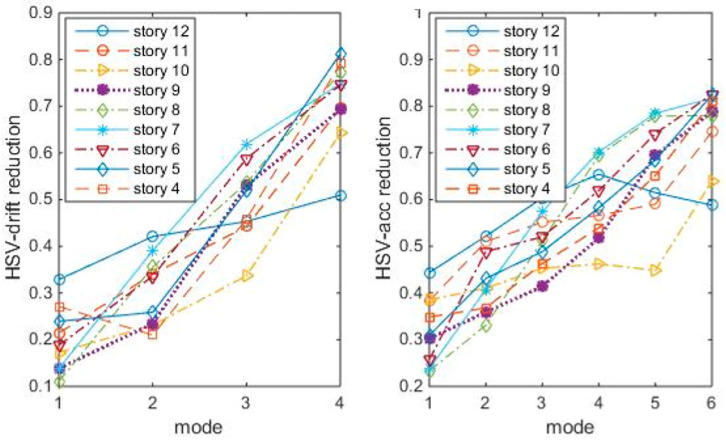
HSV reduction ratios with TMDs tuning to 2nd modes.

**Figure 7 entropy-25-01110-f007:**
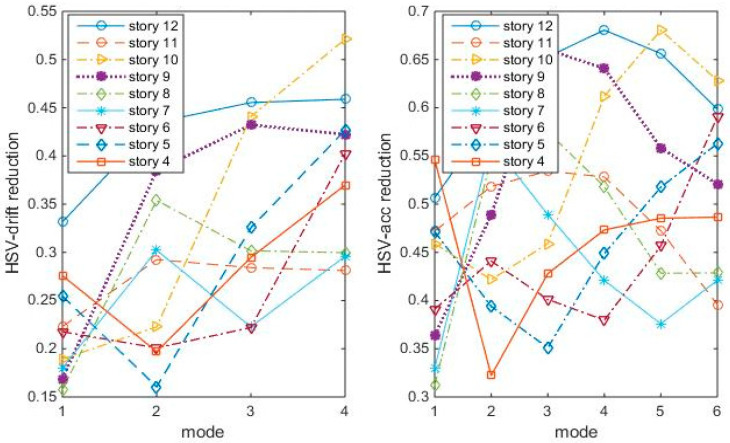
HSV reduction ratios with TMDs tuning to 3rd modes.

**Figure 8 entropy-25-01110-f008:**
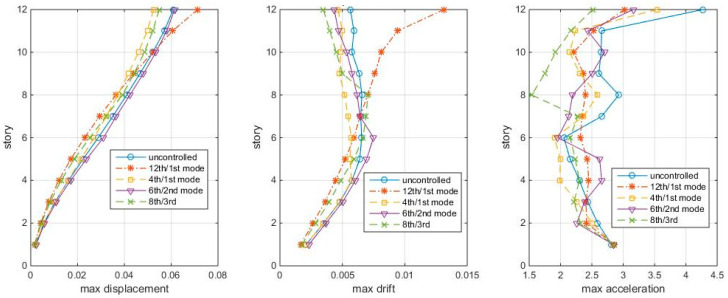
Max. Responses to white noise, with optimal TMDs (shear-bending structure).

**Figure 9 entropy-25-01110-f009:**
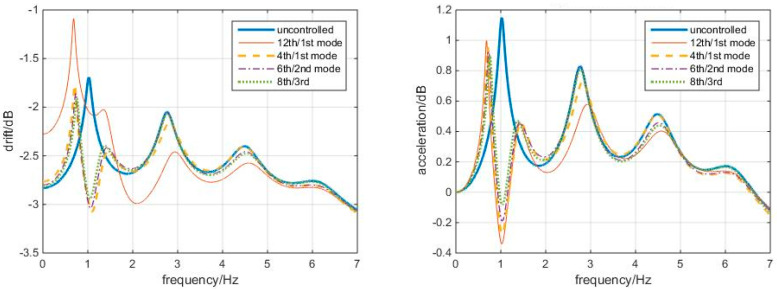
Frequency responses to white noise Δt = 0.01 s, with optimal TMDs (shear–bending structure).

**Figure 10 entropy-25-01110-f010:**
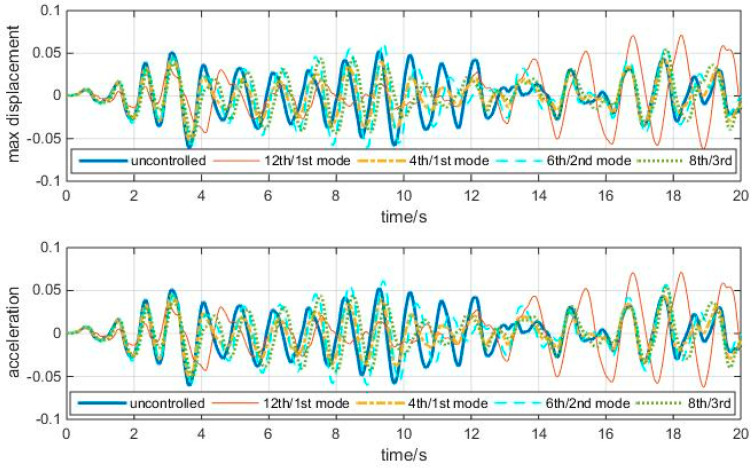
Time history responses to white noise, with optimal TMDs (shear–bending structure).

**Figure 11 entropy-25-01110-f011:**
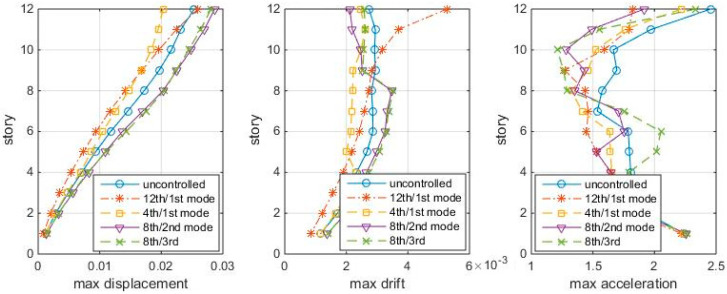
Max. responses to white noise Δt = 0.005 s, with optimal TMDs (shear structure).

**Figure 12 entropy-25-01110-f012:**
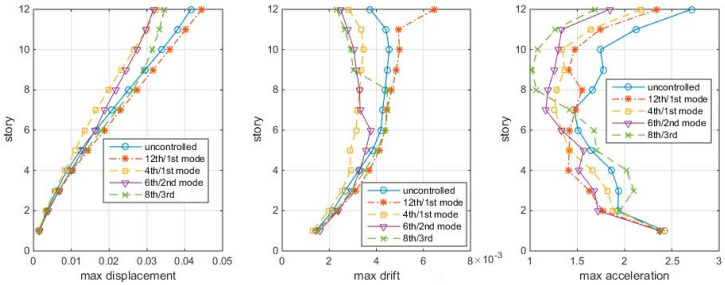
Max. responses to white noise Δt = 0.005 s, with optimal TMDs (shear–bending structure).

**Figure 13 entropy-25-01110-f013:**
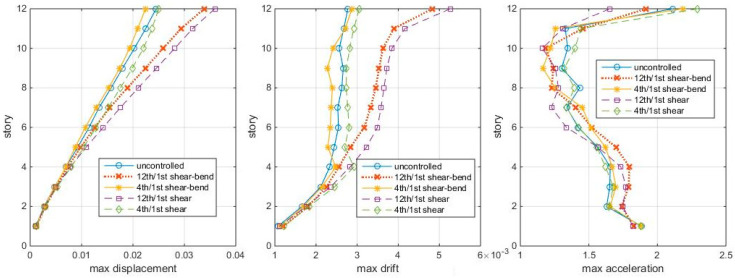
Max. responses to white noise, with TMDs’ tuning considering and not considering bending (shear–bending structure, Δt = 0.005 s).

**Figure 14 entropy-25-01110-f014:**
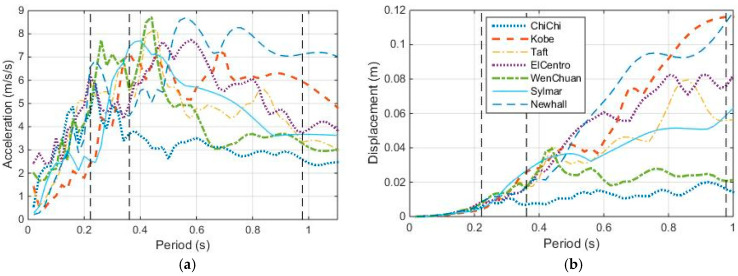
Response spectra of the seven earthquake ground motions. (**a**) Acceleration response spectra; (**b**) Displacement response spectra.

**Figure 15 entropy-25-01110-f015:**
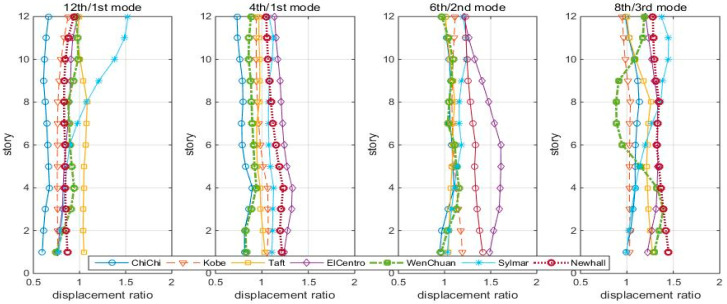
Max. Displacement reduction ratios (controlled/uncontrolled) of the shear-bending structure to seven scaled earthquakes.

**Figure 16 entropy-25-01110-f016:**
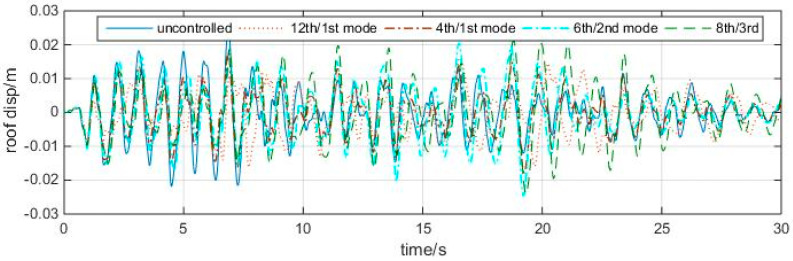
Roof displacement time history responses of the four cases to the ChiChi earthquake.

**Table 1 entropy-25-01110-t001:** Lateral stiffness from FE analysis (unit: N, m).

Stories	1	2	3	4	5	6	7	8	9	10	11	12
frame k_0_ (10^8^)	14.3	17.2	16.9	16.1	14.9	14.1	13.7	12.6	11.3	9.85	8.15	5.20
wall k_s_ (10^8^)	65.3	35.5	23.5	17.3	13.5	10.8	8.65	6.89	5.34	3.90	2.51	1.14
wall k_b_ (10^8^)	5260	2250	1420	967	701	500	353	241	155	90.6	43.8	13.7
ϕ (N.m^2^,deduced)	46.7	57.7	55.0	50.8	47.1	41.9	36.8	31.4	25.9	20.5	15.2	1.78

Note: k_0_: shear stiffness of the frame; k_s_: shear stiffness of the wall; k_b_: bending stiffness; ϕ: stiffness ratio.

**Table 2 entropy-25-01110-t002:** Peak displacement reduction ratios of roofs under seven earthquakes.

	ChiChi	Kobe	Taft	ElCentro	WenChuan	Sylmar	Newhall	Effective No.
12th/1st mode	0.6645	0.8769	0.9950	0.9815	0.9605	1.5264	0.9392	6
4th/1st mode	0.7316	0.9413	0.9686	1.1364	0.8817	1.0769	1.0461	4
6th/2nd mode	1.0118	1.1155	1.0090	1.2117	0.9698	1.1998	1.2233	1
8th/3rd mode	0.9830	0.9506	0.9972	1.2009	1.1845	1.3705	1.2800	3

**Table 3 entropy-25-01110-t003:** RMS displacement reduction of roof under seven earthquakes.

	ChiChi	Kobe	Taft	ElCentro	WenChuan	Sylmar	Newhall
12th/1st mode	0.9634	0.6987	0.6279	0.6182	0.7860	0.7610	0.5077
4th/1st mode	0.9542	0.7499	0.8208	0.9863	0.8302	0.8628	0.8835
6th/2nd mode	0.7559	0.9880	0.6631	0.8371	0.7158	0.7703	0.9781
8th/3rd mode	0.6275	0.6341	0.7082	0.8230	0.9749	0.9061	1.0288

## Data Availability

Research data are available on request from the authors.

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
