# Peer review of "TMD Design by an Entropy Index for Seismic Control of Tall Shear-Bending Buildings"

_entropy, 2023, doi:10.3390/e25081110_

Round 1

Reviewer 1 Report

The manuscript deals with the optimal design and placement of tuned mass dampers in building with considering bending effects. The method and the used model have been explained thoroughly. The results have been presented under various types of excitations and by considering different scenarios. The paper is written good, and it can be accepted for publication.

Author Response

Thank you for your recognition of this paper. Anyway, I still made some revisions as per other two reviewer's advice, please take a look and see if they are appropriate. The revisions made on abstract and introduction to polish English, delete six less relevant references, adjust some text sequences, and add some interpretations on solving procedures, etc.

Reviewer 2 Report

Unfortunately the authors do not clearly elaborate hopr their method of Hankel singular values (HSV) is used

to place the TMD in the structure used for valdation.

The validation example show results with different ? location of TMDs without adressing clearly how the situations are different and which approach has been used to define the appropriate locations.

All results focus on reduction factors for structural response under dynamic excitation. results should be condensed to relevant ones , and especially commented with regard to the proposed method of HSV for TMD location.

whole article needs fundamental english language improveent.

e.g. page 1,  liens 33,34

"One of the reasons is that this roof/1st mode scheme only has impact on

the stories near where it is located and thus is more controllable to higher modes"

or

page 2 line 51,52

"But studies specifically on bending influences on building’s seismic performance are not many. Komarizadehasl et al. [8] proposed two

method for obtaining the plastic hinge modification factors for shear wall with flexure behavior."

Author Response

point 1: The validation example show results with different ? location of TMDs without addressing clearly how the situations are different and which approach has been used to define the appropriate locations.

reply: Procedures to calculate the index and determine the appropriate locations are added under subsection 4.1, including a figure.

point 2: All results focus on reduction factors for structural response under dynamic excitation. results should be condensed to relevant ones , and especially commented with regard to the proposed method of HSV for TMD location.

reply: The objective of this study is seismic response reduction by TMD. The location and tuning determined by the entropy index have to be validated by actual structural responses. I re-wrote the abstract to clarify the purpose of the study. Also, please refer to the reply to point 1. I hope the revise (interpretation) provide more condensed and relavant theory as well as procedures.

point 3: Comments on the Quality of English Language

whole article needs fundamental English language improvement.

 e.g. page 1,  liens 33,34

"One of the reasons is that this roof/1st mode scheme only has impact on the stories near where it is located and thus is more controllable to higher modes"

 or page 2 line 51,52

"But studies specifically on bending influences on building’s seismic performance are not many. Komarizadehasl et al. [8] proposed two method for obtaining the plastic hinge modification factors for shear wall with flexure behavior."

reply: I proof-read and re-wrote the entire introduction. Hope the English and logic are OK now. I also deleted six less relavant references.

Reviewer 3 Report

This manuscript addresses the optimal configuration of Tuned Mass Dampers (TMDs) installed in a multi-story building with a hybrid shear-bending mode. The authors correctly highlight that TMDs designed for seismic engineering may not be as effective for wind applications due to the wider frequency band of seismic excitation. The paper specifically focuses on determining the optimal location of TMDs to control building response. To achieve this, the authors propose an entropy-based index that reflects the TMD configuration. Overall, the paper is well-written and makes a significant theoretical contribution. The numerical example demonstrates the applicability of the proposed entropy-based index. However, the reviewer suggests a minor revision of this manuscript before publication, offering the following comments: 

Lines 138 and 139: The reviewer could not locate matrices A and B before their appearance. Please provide their introduction or clarification before these lines. 

Equations (5) and (8) represent the equations of motion without damping, whereas the state space form in Equation (9) includes a damping matrix. Please specify the type of damping adopted in this study. 

The symbol γ appears in Equation (12) in bold form, while in Lines 166-167, it is represented in a non-bold format. Please confirm the correct formatting of this symbol. 

Is the proposed entropy-based index independent of external excitation? The optimal location of TMDs is inherently influenced by external excitation. If the proposed index is independent, please explain how it accounts for the influence of non-stationarity (both in the time and frequency domains) of a colored seismic excitation. 

Please provide clarification regarding the records/station and direction of the seismic excitation used for validation in Section 5.

Need further polish

Author Response

Point 1: Lines 138 and 139: The reviewer could not locate matrices A and B before their appearance. Please provide their introduction or clarification before these lines. 

reply: I moved the paragraph under Eqn. (9), and modified to “The influence of the location of “external” forces, such as earthquakes and active control forces, is included in the input gain, i.e., the B matrix; while TMD force is essentially “internal”, because it is the direct function of structural responses, and thus, its location impact is on the mass and /or the stiffness matrix, in all, the A matrix.”.

point 2: Equations (5) and (8) represent the equations of motion without damping, whereas the state space form in Equation (9) includes a damping matrix. Please specify the type of damping adopted in this study. 

reply: The damping term is added to Eqn.(5), and a paragraph is added under Eqn.(5) to explain the damping used in this study, and why it is neglected in the condensation deduction.

point 3: The symbol γ appears in Equation (12) in bold form, while in Lines 166-167, it is represented in a non-bold format. Please confirm the correct formatting of this symbol. 

reply: The symbol is revised to Bold.

point 4: Is the proposed entropy-based index independent of external excitation? The optimal location of TMDs is inherently influenced by external excitation. If the proposed index is independent, please explain how it accounts for the influence of non-stationarity (both in the time and frequency domains) of a colored seismic excitation. 

reply: I added a few sentence under the subtitle 4.2 about the necessity of examining the white noice, and modified the paragraph under Fig. 11 as: “different “optimal” TMDs correspond to ground excitations with different dominant frequencies. However, the entropy-selected TMDs all lead to mitigated seismic responses than the uncontrolled case. The fact shows that, design a passive controller for non-stationary or colored excitations, using the system inherent property as index is not a bad choice.”

I also re-wrote the abstract to make it clearer. There are also other revise as per other reviewer's advice. Please take a look and see if they are appropriate.

point 5: Please provide clarification regarding the records/station and direction of the seismic excitation used for validation in Section 5.

reply: A short interpretation about the earthquake records was added.

point 6: English needs further polish

reply: I re-wrote the abstract and revised the introduction and some other texts to polish English. Hope it is better now.

Thank you for your keen advice!

Round 2

Reviewer 3 Report

The authors respond to my comments properly. The manuscript can be accepted in the present form. 

Minor editing of English language required. It is not mandatory. 

Author Response

Thank you for your rigorous academic attitude. I would try my best to eliminate the bug in my English... 

  1. The last sentence in conclusion is revised.
  2. The abstract is revised.
  3. I also re-arranged the order of a few sentences in the introduction, and accordingly, the reference number.
  4. There are a few other minor changes in English.

Please do not hesitate to point out other errors. I would be glad to learn and make a progress in writing.
